# Radar Target Localization with Multipath Exploitation in Dense Clutter Environments

Rui Ding , Zhuang Wang *, Libing Jiang and Shuyu Zheng

National Key Laboratory of Science and Technology on Automatic Target Recognition, National University of Defense Technology, Changsha 410072, China
* Correspondence: zhuang_wang@sina.com

**Abstract:** The performance of classic radar geometry based on the line-of-sight (LOS) signal transmitted from radar to the target in the free space is affected by multipath echoes in urban areas, where non-line-of-sight (NLOS) signals reflected by obstacles are received by the radar. Based on prior information of the urban situation, this article proposes a novel two-stage localization algorithm with multipath exploitation in a dense clutter environment. In the offline stage, multipath propagation parameters of uniformly distributed samples in the radar field of view are predicted by the ray-tracing technique. In the online stage, a rough location of the target is estimated by the maximum similarity between measurements and the predicted parameters of reference samples at different locations. The similarity is described by the likelihood between measurements and the predicted multipath parameters with respect to all possible associated hypotheses. A gating threshold is derived to exclude less likely hypotheses and reduce the computational burden. The accurate target location is acquired by a non-linear least squares (NLS) optimization of the associated multipath components. Simulation results in various noise conditions show that the proposed method provides robust and accurate target localization results under dense clutter conditions, and the offline pre-calculation of ray-tracing ensures the real-time performance of the proposed localization algorithm. The root mean square error (RMSE) of simulation results shows the advantage of the proposed method over the existing method. The presented results suggest that the proposed method can be applied to NLOS target localization applications in complex environments.

**Keywords:** multipath exploitation; non-line-of-sight (NLOS); target localization; data association; non-linear least squares (NLS) localization; dense clutter



## 1. Introduction

Target localization in urban areas is an important requirement for both military and civilian applications, such as emergency rescue, security surveillance and intelligent transportation. A variety of techniques has been developed to detect and locate targets, including vision cameras, sonars, lidars and radars [1]. The all-time and all-weather advantages make radar an irreplaceable part of these techniques. However, dense buildings become obstacles to radar electromagnetic (EM) waves, which make EM waves propagate through not only the line-of-sight (LOS) path but also non-line-of-sight (NLOS) paths, such as reflection, diffraction and transmission. Radar systems commonly acquire range and angle measurements from time-of-arrival (TOA) [2,3] and direction-of-arrival (DOA) [4,5] of echo signals as LOS path returns. However, the lengths of received NLOS signals are longer than the range of the direct path from radar to the target in multipath propagations, and the received DOA of NLOS signals are different from the DOA of LOS path. Therefore the NLOS propagations lead to considerable errors in range and angle measurements in multipath scenarios [6]. The performances of classic radar target localization methods based on LOS geometry of free space propagation degrade significantly in such multipath environments.

To mitigate the multipath effect, multiple techniques have been proposed. Conventional researchers consider multipath signals as clutter and attempt to separate multipath

signals from the received signal so that only the LOS signal is applied for further processing [7,8]. However, multipath signals propagated through NLOS paths provide extra spatial information that can be utilized to detect and locate targets. Early studies in constrained multipath scenarios demonstrated the possibility and effectiveness of multipath exploitation where the fixed multipath propagation can be modeled via analytical geometry methods [9–12]. A typical application is the over-the-horizon radar (OHTR) where EM waves are reflected by ionosphere layers at different heights. Classic data association methods (e.g., joint probabilistic data association filter (JPDAF), probabilistic multi-hypothesis tracker (PMHT) and probability hypothesis density (PHD) filter) combined the analytical geometry reflection model with unknown height have been proposed to improve the performance of OHTR target tracking problem [13–17]. Through-the-wall radar is another application that exploits the transmission of EM waves to detect and locate targets behind the wall [18–20].

In urban environments, a common task is indoor positioning which locates a target in the presence of both LOS and NLOS signals [21,22]. In this case, the NLOS signal reflected by obstacle walls is equivalent to the bistatic radar geometry model where a signal is transmitted from the radar tothe target and received by a virtual radar mirrored on the reflecting wall with the original radar. Therefore, target localization can be derived from the LOS path combined with the NLOS paths using the bistatic radar geometry without extra radar sources. In [23], the propagation path and wall association algorithm is proposed to associate multipath TOA signals to their corresponding reflecting obstacles, and target localization is derived by a non-linear least squares (NLS) optimization. The Cramer–Rao bound of the TOA-based multipath localization method is given in [24] to analyze the localization performance of radar at different places and obtain the optimal radar position setup in the given scenario. In [25], directional constraints are introduced to ultra-wideband radar for indoor positioning tasks to improve location accuracy in the absence of some NLOS paths. Liu et al. [26] proposed a multi-target location method based on target association hypothesis and consistency checking. Another typical task in urban areas is to detect targets behind the corner, where the LOS path does not exist and only reflected signals can be obtained [27,28]. Considering at least two NLOS paths are required to locate targets behind the corner, the multipath returns with the two largest echo amplitudes are selected to obtain candidate target positions, and the target true position is further derived by a correlation coefficient function [29,30]. Researchers also attempted to solve the NLOS target localization problem under real-world conditions with the latest deep learning methods [31–33].

The studies above based on analytical geometry modeling of multipath propagation implied prior information of the propagation model determined by the specific scenario, while the propagation model mismatching leads to significant localization error in the actual complex scene, especially in dense clutter environments. A widely used solution to predict EM wave propagation characteristics precisely is the ray-tracing technique based on physical optics (PO) which approximates the EM wave propagation in terms of rays [34,35]. The reflection, diffraction and transmission paths from the radar to the target can be computed with known scene information. A sequential Monte Carlo PHD filter based on real-time ray-tracing prediction is implemented to track mobile terminals in a large-scale urban scene [36]. For simple scenarios, validation of a 2D ray-tracer has been proved by comparison with measured data [37]. The ray-tracing technique is also used to generate multipath fingerprints for localization methods based on DOA, time difference of arrival or received signal strength [38–40]. However, real-time ray-tracing calculation is time-consuming in complex environments because of its high computational complexity. This paper concentrates on a preliminary offline calculation for ray-tracing prediction of multipath signals combined with high-accuracy real-time online localization in dense clutter environments. The main contributions of this paper are summarised as follows:

1. Derivation of the likelihood of cluttered multipath measurement with respect to the reference multipath parameters predicted by the ray-tracing technique and the approximation of multipath likelihood;
2. Proposal of an accurate target localization method based on NLS optimizations of associated multipath measurements;
3. Simulation results in various conditions validating the robustness and performance of the proposed method.

The remainder of this paper is organized as follows. Section 2 introduces the multipath propagation model and the multipath measurement model used in this paper. Section 3 describes the proposed localization algorithm in detail. In Section 4, simulations are carried out in different conditions to analyze the performance of the proposed method. Finally, conclusions are given in Section 5.

## 2. Model

In this section, we describe the multipath propagation model generated through the ray-tracing technique and the measurement model used by the proposed algorithm in Section 3.

### 2.1. Multipath Propagation Model via Ray-Tracing

Computational electromagnetics is widely used to simulate the propagation of EM waves in the field of communication, chip design and radar. The ray-tracing technique provides an effective approximation of EM waves when the wavelength is small compared to the size of structures in large-scale scenes, where the full-wave numerical method, such as a finite-difference time-domain (FDTD) method, cannot be applied because of the high computational complexity. The ray-tracing method describes the reflection, diffraction, and transmission of EM waves based on PO theory by representing EM waves radiated from the source as rays [41]. Since the implementation of the ray-tracing method is beyond the scope of our paper, details of the ray-tracing technique are not described further. In this paper, reflection paths from a given radar position to the target position are calculated by a commercial software Wireless Insite [42], with environmental information supposed to be prior knowledge. The diffraction paths are not included in the simulation because the irregular propagation geometry of diffraction paths is usually not applicable to localization algorithms. Let us denote the radar position by

$$\mathbf{r} = [x_r, y_r]^T, \tag{1}$$

and the target position is

$$\mathbf{x} = [x_d, y_d]^T. \tag{2}$$

Then the detections of multipath propagation parameters can be abstracted into a non-linear function related to radar and target position as

$$\mathbf{H}(\mathbf{r}, \mathbf{x}) = \mathbf{H_x} = \{\mathbf{h_x}^k\}_{k=1}^{N_p}, \tag{3}$$

where $N_p$ is the number of propagation paths. and $\mathbf{h_x}$ is the measurement vector for each path composed of corresponding TOA $\tau$ and DOA $\phi$ given by

$$\mathbf{h_x}^k = [\tau_\mathbf{x}^k, \phi_\mathbf{x}^k]^T. \tag{4}$$

Note that $N_p$ varies when the target location changes, so the set representation $\mathbf{H_x}$ is used instead of vector representation, indicating that the number of elements is not fixed.

Since the multipath propagation $\mathbf{H}(\mathbf{r}, \mathbf{x})$ has no closed form, an intuitive idea is to sample $\mathbf{H_x}$ with a numerical method. Randomly distributed particles on the target state space sampled by a real-time ray-tracing method in previous studies revealed high-precision positioning results, while real-time ray-tracing calculation is time-consuming for the high

computational complexity [43]. Instead, this paper uses fixed uniformly distributed samples which can be computed offline in advance. The position of uniformly distributed samples are indexed as

$$\mathbf{s}(i,j) = \mathbf{s}_{ij} = [i \cdot \Delta x, j \cdot \Delta y]^T, 1 \le i \le N_x, 1 \le j \le N_y, \tag{5}$$

where $\Delta x$ and $\Delta y$ are the interval of samples in the x and y direction, and $N_x$ and $N_y$ are the corresponding number of samples. Figure 1 illustrates an example of a 3D ray-tracing result from Wireless Insite software and a 2D simplification ray-tracing result which ignored the altitude in elevation. For ease of simulation and illustration, the paper makes some simplifications, and 2D ray-tracing is adopted for subsequent simulation verification. Meanwhile, the method can be extended easily to 3D scenarios without modifying the localization algorithm. The parameters of single-trip multipath propagation from the radar to the sampled target positions are acquired by a ray-tracing method with proper simulation setups as

$$\mathbf{H}_1(i,j) = \{\mathbf{h}_1(i,j)\}_{k_1=1}^{N_1}, \tag{6}$$

where $N_1$ is the number of single-trip paths. The round-trip path from the radar to the target and back to the radar is a combination of any two single-trip paths, so the number of round-trip paths is given as

$$N_p = N_1^2. \tag{7}$$

The TOA of the round-trip path is the sum of two single-trip paths, and the DOA of the round-trip path is equal to the DOA of the return path; thus, the multipath parameter set of round-trip propagation for the target located at $\mathbf{x}(i,j)$ is given as

$$\mathbf{H}_{ij} = \{\mathbf{h}_{i,j}^k\}_{k=1}^{N_p}, \tag{8}$$

where each measurement vector is given by

$$\mathbf{h}_{i,j}^k = [\tau_{ij}^{k_1} + \tau_{ij}^{k_2}, \phi_{ij}^{k_2}], \quad 1 \le k_1, k_2 \le N_l. \tag{9}$$

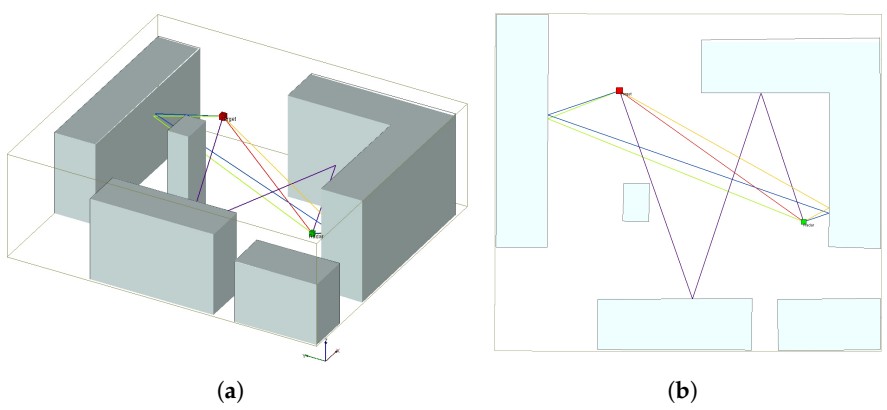

|     |     |
| :-: | :-: |
| (**a**) | (**b**) |

**Figure 1.** Ray-tracing simulation examples: (**a**) 3D ray-tracing; (**b**) 2D ray-tracing.

### 2.2. Measurement Model

In this subsection, we will introduce the radar measurement model. Let us denote the measured set of multipath echo as

$$\mathbf{Z} = \{\mathbf{z}^j\}_{j=1}^{N_m}, \tag{10}$$

where $N_m$ is the number of detections, and $\mathbf{z}^j$ is the measurement vector composed of TOA and DOA pairs as

$$\mathbf{z}^j = [\tau^j, \phi^j]^T. \tag{11}$$

The radar detection process will produce false alarms and noise; therefore, the measured set $\mathbf{Z}$ is not the same as the theoretical multipath parameter set $\mathbf{H_x}$ of the target. The measurement set consists of two parts denoted by

$$\mathbf{Z} = \mathbf{C} \cup \mathbf{Z_x}, \tag{12}$$

where the first part $\mathbf{C}$ denotes false alarms generated from clutter, and the second part $\mathbf{Z_x}$ denotes measurements from the detected multipath components of target $\mathbf{H_x}$.

The clutter $\mathbf{C}$ is modeled as a Poisson point process (PPP) with Poisson rate $\lambda$; thus, the number of clutter measurements is a Poisson distribution with the probability density function (PDF) given by

$$p(|\mathbf{C}| = n) = e^{-\lambda}\frac{\lambda^n}{n!}, n = 0, 1, \cdots \tag{13}$$

Each clutter measurement $\mathbf{c} \in \mathbf{C}$ is randomly distributed in the measurement space independently with the PDF given by

$$p(\mathbf{c}) = \frac{1}{V}, \tag{14}$$

where $V$ is the volume of the measurement space. The measurement vector $\mathbf{z}$ is specified by the radar TOA and DOA range as

$$\tau \in [\tau_{min}, \tau_{max}], \phi \in [\phi_{min}, \phi_{max}], \tag{15}$$

thus the volume $V$ is derived as

$$V = (\tau_{max} - \tau_{min}) \cdot (\phi_{max} - \phi_{min}). \tag{16}$$

The second part $\mathbf{Z_x}$ is detections from the target. Each path of multipath echos from the target either generates one measurement with detection probability $P_D$, or becomes a missed detection with probability $1 - P_D$. The measurement noise for each detected path is modeled as an additive zero-mean Gaussian white noise $\mathbf{w}$; therefore the detected set $\mathbf{Z_x}$ is the union of detections of each path expressed as

$$\mathbf{Z_x} = D(\mathbf{h_x^1} + \mathbf{w}) \cup \cdots \cup D(\mathbf{h_x^{N_p}} + \mathbf{w}), \tag{17}$$

where $D(\cdot)$ denotes for the detection process which is a Bernoulli distribution with probability parameter $P_D$ denoted as

$$\begin{cases} Pr(D(\mathbf{h_x} + \mathbf{w}) = \{\mathbf{h_x} + \mathbf{w}\}) = P_D, \\ Pr(D(\mathbf{h_x} + \mathbf{w}) = \varnothing) = 1 - P_D. \end{cases} \tag{18}$$

## 3. Method

In this section, the proposed two-stage localization algorithm is introduced. Figure 2 illustrates the framework of target detection and the localization algorithm. The overall structure of the proposed algorithm contains an offline stage and an online stage. In the offline stage, multipath propagation parameters of uniformly distributed targets are simulated by the ray-tracing technique introduced in Section 2.1. In the online stage, a two-step localization algorithm with multipath exploitation is proposed to locate targets with dense clutter. The first step is to obtain the multipath propagation model and a rough location by the maximum likelihood estimation (MLE) of measurement set $\mathbf{Z}$ and the sampled target multipath parameters $\mathbf{H}_{ij}$. In the second step, the accurate target location is estimated by an NLS method to eliminate off-grid errors originating from samples. A detailed discussion of the proposed likelihood function and localization algorithm is presented in the following subsections.

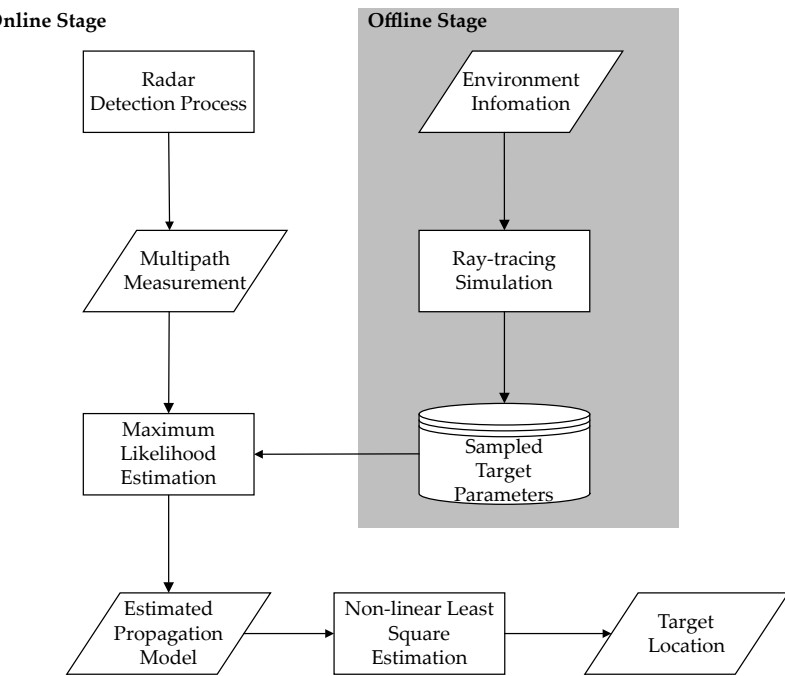

**Figure 2.** Framework of the proposed target localization algorithm.

### 3.1. Likelihood Function with Multi-Measurement

The classic likelihood function describes the probability between target state and measurement, assuming that a target generates at most one measurement in one observation. However, this is not the case in multipath scenarios where a target may generate more than one measurement. Therefore, finite-set statistics (FISST) theory [44] is introduced to describe both the uncertainty of target location and the uncertainty of multipath propagation simultaneously. This subsection derives the likelihood function $f(\mathbf{Z}|\mathbf{H_x})$ to describe the conditional probability density that the measurement set $\mathbf{Z}$ is generated by a potential target multipath propagation $\mathbf{H_x}$.

We introduce a multipath measurement association hypothesis variable $\boldsymbol{\theta} = [\theta_1, \cdots, \theta_{N_p}]$ to interpret whether each path $\mathbf{h_x}^k$ is detected in the measurement set $\mathbf{Z}$ as

$$\theta_k = \begin{cases} j \in \{1, \cdots, N_m\}, & \text{if } \mathbf{z}^j \text{ is the detection of } \mathbf{h_x}^k, \\ 0, & \text{if } \mathbf{h_x}^k \text{ is not detected.} \end{cases} \tag{19}$$

So the measurements generated from multipath component detections are given as

$$\mathbf{z_x}^{\theta_k} = \mathbf{h_x}^k + \mathbf{w}, k = 1, \cdots, N_p. \tag{20}$$

With the assumption that all measurements, including false alarms from clutter and detections from target multipath returns, are independent of each other, the conditional probability density of detected set $\mathbf{Z_x}$ with the given association hypothesis is derived as

$$p(\mathbf{Z_x}|\mathbf{H_x}, \boldsymbol{\theta}) = \prod_{\theta_k=0} (1 - P_d) \prod_{\theta_k>0} P_d \cdot p(\mathbf{z}^{\theta_k}|\mathbf{h_x}^k), \tag{21}$$

where the first product component denotes the probability of undetected paths, and the second component denotes the probability of detected paths, and the conditional probability density of each measurement given the associated path parameter is modeled as a Gaussian distribution

$$p(\mathbf{z}|\mathbf{h_x}) = \frac{1}{(2\pi)^{\frac{N}{2}} \det^{\frac{1}{2}}(\boldsymbol{\Sigma})} \exp\left[-\frac{1}{2}(\mathbf{z} - \mathbf{h_x})^T \boldsymbol{\Sigma}^{-1}(\mathbf{z} - \mathbf{h_x})\right], \tag{22}$$

where $N$ is the dimension of vector $\mathbf{h_x}$, and $\mathbf{\Sigma}$ is the covariance of measurement noise $\mathbf{w}$.

Measurements unassociated with target multipath echos in the hypothesis are false alarms generated from clutter, which can be expressed as

$$\mathbf{C} = \{\mathbf{z}^j | \nexists \theta_k = j, k = 1, \cdots, N_p\}. \tag{23}$$

Then, the probability function of clutter conditioned on an association hypothesis is given as

$$p_c(\mathbf{C}|\boldsymbol{\theta}) = e^{-\lambda}(\frac{\lambda}{V})^{|\mathbf{C}|}, \tag{24}$$

where $|C|$ denotes the cardinality of the set.

The measurement likelihood conditioned on an association hypothesis is given as

$$p(\mathbf{Z}|\mathbf{H_x}, \boldsymbol{\theta}) = p_c(\mathbf{C}|\boldsymbol{\theta})p(\mathbf{Z_x}|\mathbf{H_x}, \boldsymbol{\theta}). \tag{25}$$

Finally, the likelihood function of measurement set $\mathbf{Z}$ given target state set $\mathbf{H_x}$ is derived by summing up all association hypotheses as

$$p(\mathbf{Z}|\mathbf{H_x}) = \sum_{\boldsymbol{\theta} \in \boldsymbol{\Theta}} p(\mathbf{Z}|\mathbf{H_x}, \boldsymbol{\theta}), \tag{26}$$

where $\boldsymbol{\Theta}$ denotes the space of all possible association hypotheses.

### 3.2. Approximation of Likelihood

Let us consider the number of all possible hypotheses in $\boldsymbol{\Theta}$. The association variable $\boldsymbol{\theta}$ is an $N_p$-dimensional vector; therefore, the number of nonzero values of $\boldsymbol{\theta}$ does not exceed $N_p$. In addition, each measurement $\mathbf{z}$ in the measurement set is associated with the target propagation path at most once; thus, the number of nonzero values of $\boldsymbol{\theta}$ does not exceed $N_m$. Considering the case where $\boldsymbol{\theta}$ contains $k$ nonzero values, then the number of associations is expressed as

$$\binom{N_m}{k}\binom{N_p}{k}k!, \tag{27}$$

where $\binom{N_m}{k}$ denotes the number of ways to select $k$ measurements from $\mathbf{Z}$, $\binom{N_p}{k}$ denotes the number of ways to select $k$ paths from $\mathbf{H_x}$, and $k!$ denotes the number of ways to assign the selected $k$ elements. Therefore, the number of hypotheses in $\boldsymbol{\Theta}$ is given as

$$N(N_m, N_p) = \sum_{k=0}^{min(N_m, N_p)} \binom{N_m}{k}\binom{N_p}{k}k!. \tag{28}$$

Since $N(N_m, N_p)$ increases exponentially with the number of measurements and target paths, the $O(n!)$ level time complexity of $p(\mathbf{Z}|\mathbf{H_x})$ is practically impossible to compute. Hence, appropriate approximation and complexity reduction techniques are introduced to reduce the numerical burden.

No threshold is set in the previous formula indicating that the measurement can be associated with any target path parameter $\mathbf{h_x}$. However, when the measurement is far away from $\mathbf{h_x}$, the conditional probability density $p(\mathbf{z}|\mathbf{h_x})$ becomes small and can be ignored. Hence, we introduce a distance matrix $\mathbf{D}$ to describe the distance between each predicted path and measurement, where the normalized scale-invariant metric Mahalanobis distance is applied to obtain the distance between the $k$-th predicted path and $j$-th measurement as

$$d_{j,k} = \sqrt{(\mathbf{z}^j - \mathbf{h_x}^k)^T \mathbf{\Sigma}^{-1}(\mathbf{z}^j - \mathbf{h_x}^k)} \tag{29}$$

Then, only measurements within a gate around the predicted target path are considered valid, and the association matrix **G** is acquired by a gating threshold $T$ as

$$
G_{j,k} = \begin{cases} 1, & \text{if } d_{j,k}^2 < T, \\ 0, & \text{otherwise.} \end{cases} \tag{30}
$$

Without losing generality, we consider two hypotheses $\boldsymbol{\theta}$ and $\boldsymbol{\theta}'$ that differ only in the $K$-th association expressed as

$$
\begin{cases} \theta_k = \theta_k', & k = 1, \cdots, N_p, k \neq K, \\ \theta_k > 0, \theta_k' = 0, & k = K. \end{cases} \tag{31}
$$

The threshold $T$ should satisfy that the probability of a valid association between $\mathbf{z}^{\theta_K}$ and $\mathbf{h}_{\mathbf{x}}^K$ is larger than the probability that they are not associated with; therefore, we obtain the inequality as

$$
p(\mathbf{Z_x}|\mathbf{H_x}, \boldsymbol{\theta}) > p(\mathbf{Z_x}|\mathbf{H_x}, \boldsymbol{\theta}'). \tag{32}
$$

By substituting (21) and (24) into (32), we obtain

$$
P_d \cdot p(\mathbf{z}^{\theta_K}|\mathbf{h}_{\mathbf{x}}^K) > \frac{\lambda}{V}(1 - P_d). \tag{33}
$$

Therefore, the threshold $T$ is given by substituting (22) as

$$
T = -2\log\left(2\pi\sqrt{\det(\boldsymbol{\Sigma})}\frac{\lambda}{V}\frac{1 - P_d}{P_d}\right). \tag{34}
$$

The number of hypotheses decreases significantly after the gating step. All valid hypotheses are enumerated by the recursive method proposed in the next step. Considering the case where two multipath components are associated with three measurements by an association matrix

$$
\mathbf{G} = \begin{bmatrix} 1 & 1 \\ 0 & 1 \\ 0 & 0 \end{bmatrix}, \tag{35}
$$

we extend the matrix by appending an $N_m$-dimensional identity matrix to the right as

$$
\mathbf{G}' = \left[\begin{array}{cc|ccc} 1 & 1 & 1 & 0 & 0 \\ 0 & 1 & 0 & 1 & 0 \\ 0 & 0 & 0 & 0 & 1 \end{array}\right], \tag{36}
$$

where the left part denotes the associations with the target multipath, and the right part denotes the associations with clutter.

A valid assignment hypothesis under the extended association matrix expression is defined as a $N_m \times (N_p + N_m)$ matrix **A** with the elements satisfying

$$
\begin{cases} A_{jk} \in \{0,1\}, & \forall j, k, \\ \sum_j A_{jk} = 1, & \forall k, \\ \sum_k A_{jk} \in \{0,1\}, & \forall j. \end{cases} \tag{37}
$$

The first constraint denotes that the $j$-th measurement and the $k$-th target path are assigned. The second constraint denotes that each measurement must be assigned to a target path or clutter. The third constraint denotes that each target path or clutter is assigned at most once.

Considering that **G**′ is a sparse matrix, it is feasible to enumerate all possible assignment matrices **A** by a recursive enumeration method. Assume that $m$ nonzero values are found in the first row of **G**′. We have $m$ valid assignment matrices initialized with $A_{1K_1} = 1$

separately, where $K_1$ is the corresponding column of the nonzero value. We then remove the first row and the $K_1$-th column in $\mathbf{G}'$, and $\mathbf{G}'$ is reduced to a $(N_m - 1) \times (N_p + N_m - 1)$ association matrix. The procedure is repeated with the search and remove steps in the reduced association matrix until it is empty. The column of the nonzero value in the $j$-th recursion is recorded as $K_j$, therefore $A_{jK_j}$ is set to 1 recursively. The result of all valid assignment matrices for the example association matrix in (36) is given by

$$
\begin{aligned}
\mathbf{A}^1 &= \begin{bmatrix} 1 & 0 & 0 & 0 & 0 \\ 0 & 1 & 0 & 0 & 0 \\ 0 & 0 & 0 & 0 & 1 \end{bmatrix}, \\
\mathbf{A}^2 &= \begin{bmatrix} 1 & 0 & 0 & 0 & 0 \\ 0 & 0 & 0 & 1 & 0 \\ 0 & 0 & 0 & 0 & 1 \end{bmatrix}, \\
\mathbf{A}^3 &= \begin{bmatrix} 0 & 1 & 0 & 0 & 0 \\ 0 & 0 & 0 & 1 & 0 \\ 0 & 0 & 0 & 0 & 1 \end{bmatrix}, \\
\mathbf{A}^4 &= \begin{bmatrix} 0 & 0 & 1 & 0 & 0 \\ 0 & 1 & 0 & 0 & 0 \\ 0 & 0 & 0 & 0 & 1 \end{bmatrix}, \\
\mathbf{A}^5 &= \begin{bmatrix} 0 & 0 & 1 & 0 & 0 \\ 0 & 0 & 0 & 1 & 0 \\ 0 & 0 & 0 & 0 & 1 \end{bmatrix},
\end{aligned}
\tag{38}
$$

where each assignment matrix can be converted into an association vector as

$$
\theta_k = \begin{cases} j, & \exists A_{jk} = 1, k = [1, \cdots, N_p], \\ 0, & \text{otherwise.} \end{cases}
\tag{39}
$$

Therefore, the corresponding valid association vectors are given as

$$
\begin{aligned}
\boldsymbol{\theta}^1 &= [1, 2], \\
\boldsymbol{\theta}^2 &= [1, 0], \\
\boldsymbol{\theta}^3 &= [2, 0], \\
\boldsymbol{\theta}^4 &= [0, 2], \\
\boldsymbol{\theta}^5 &= [0, 0].
\end{aligned}
\tag{40}
$$

### 3.3. Target Localization Algorithm

In this subsection, we proposed a two-step localization algorithm based on the likelihood function derived in the previous subsection. In the first step, the target multipath propagation model is estimated by traversal of the maximum likelihood between measurement set $\mathbf{Z}$ and all predicted samples $\mathbf{H}_{ij}$ as

$$
\hat{\mathbf{H}} = \underset{1 \le i \le N_x, 1 \le j \le N_y}{\arg \max} \, p(\mathbf{Z}|\mathbf{H}_{ij}).
\tag{41}
$$

The sample location corresponding to the multipath propagation model $\hat{\mathbf{H}}$ obtained by MLE can be regarded as a rough estimation of the target location, while the uniformly distributed samples lead to off-grid errors. Hence, in the second step, an NLS-based localization algorithm is proposed to obtain the accurate target location based on the roughly estimated target location $\hat{\mathbf{x}}$.

By law of reflection, the reflection path from radar to the target is equivalent to the path propagated to the target from a virtual radar mirroring the real radar with respect to the

reflection obstacle as illustrated in Figure 3. We denote the normal vector of the reflecting surface as $\mathbf{n} = [n_1, n_2, n_3]^T$; therefore, the reflecting plane equation can be expressed as

$$\mathbf{n}^T \cdot [x - x_0, y - y_0, z - z_0]^T = 0, \tag{42}$$

where $\mathbf{t} = [x_0, y_0, z_0]^T$ is a point on the plane. Then, the distance from the radar $\mathbf{r} = [x_r, y_r, z_r]^T$ to the plane is given as

$$d = \frac{\mathbf{n}^T(\mathbf{r} - \mathbf{t})}{\|\mathbf{n}\|}, \tag{43}$$

where $\|\mathbf{n}\| = \sqrt{n_1^2 + n_2^2 + n_3^2}$ is the length of the vector. The virtual radar position with the same distance as $\mathbf{r}$ in the opposite direction is derived as

$$\mathbf{r}^1 = \mathbf{r} - 2d\frac{\mathbf{n}}{\|\mathbf{n}\|}. \tag{44}$$

We assume that $\mathbf{n}$ is a unit normal vector; thus, the mirrored virtual radar position can be rewritten into a matrix form as

$$\begin{bmatrix} x_{r_1} \\ y_{r_1} \\ z_{r_1} \end{bmatrix} = \begin{bmatrix} 1 - 2n_1 n_1 & -2n_2 n_1 & -2n_3 n_1 \\ -2n_1 n_2 & 1 - 2n_2 n_2 & -2n_3 n_2 \\ -2n_1 n_3 & -2n_2 n_3 & 1 - 2n_3 n_3 \end{bmatrix} \begin{bmatrix} x_r \\ y_r \\ z_r \end{bmatrix} + 2\mathbf{n}^T\mathbf{t} \begin{bmatrix} n_1 \\ n_2 \\ n_3 \end{bmatrix}. \tag{45}$$

Therefore, the mirroring of a radar point $\mathbf{r}$ with respect to a plane $p$ is a linear transformation denoted as

$$\mathbf{r}^1 = \mathbf{S}_p(\mathbf{r}). \tag{46}$$

The virtual radar position of multi-bounce reflection can be derived by the cascade of multiple single-bounce reflections. Let us denote the single-trip multi-bounce reflection path from target to the target as $R - O_1 - \cdots - O_n - T$, where $O_1, \cdots, O_n$ are reflection points with corresponding reflection planes $p_1, \cdots, p_n$. The virtual radar with respect to $p_1$ is given as $\mathbf{r}^1 = \mathbf{S}_{p_1}(\mathbf{r})$, and the equivalent path starting from virtual radar is given as $R_1 - O_2 - \cdots - O_n - T$. By recursion, the equivalent virtual radar of the single-trip multi-bounce reflection is obtained by

$$\mathbf{r}^n = \mathbf{S}_{p_n}(\mathbf{r}^{n-1}) = \mathbf{S}_{p_n}(\mathbf{S}_{p_{n-1}}(\mathbf{r}^{n-2}))\cdots \equiv \mathbf{S}(\mathbf{r}). \tag{47}$$

As described in the previous subsection, measurements and predicted paths of $\hat{\mathbf{x}}$ are associated by the assignment matrices. Let us denote one of the predicted paths $\mathbf{p}_{\hat{\mathbf{x}}}$ is associated with measurement $\mathbf{z}$. The round-trip signal received by the radar consists of single-trip paths, the transmitted path and the received path. The two virtual radars $\mathbf{r}_1 = [x_{r_1}, y_{r_1}]^T$ and $\mathbf{r}_2 = [x_{r_2}, y_{r_2}]^T$ of the transmitted path and the received path can be obtained by (47). If the transmitted path and the received path are the same, the radar geometry is equivalent to a monostatic radar geometry from virtual radar to the target directly. If the two paths are different, the radar geometry is equivalent to a bistatic radar geometry where the signal is transmitted from virtual radar $\mathbf{r}_1$ to the target and received by virtual radar $\mathbf{r}_2$. Therefore, the length of the received round-trip path is derived as

$$h_1 = \sqrt{(x_{r_1} - x_d)^2 + (y_{r_1} - y_d)^2} + \sqrt{(x_{r_2} - x_d)^2 + (y_{r_2} - y_d)^2}, \tag{48}$$

where $\mathbf{x} = [x_d, y_d]^T$ is the target location, and the received DOA is given as

$$h_2 = \arcsin\left( \frac{y_{r_2} - y_d}{\sqrt{(x_{r_2} - x_d)^2 + (y_{r_2} - y_d)^2}} \right). \tag{49}$$

The path parameter is expressed as a non-linear function of the target location by

$$\mathbf{h}(\mathbf{x}) = [h_1, h_2]^T. \tag{50}$$

The conditional probability function of the measurement $\mathbf{z}$ with respect to the associated path is given as

$$p(\mathbf{z}|\mathbf{h}(\mathbf{x})) = \frac{1}{2\pi \det^{\frac{1}{2}}(\boldsymbol{\Sigma})} \exp\left[-\frac{1}{2}(\mathbf{z} - \mathbf{h}(\mathbf{x}))^T \boldsymbol{\Sigma}^{-1}(\mathbf{z} - \mathbf{h}(\mathbf{x}))\right], \tag{51}$$

where $\mathbf{h} = [h_1, h_2]^T$. Therefore, the probability of the assignment $\hat{\theta}$ is derived as

$$p(\mathbf{Z}|\mathbf{x}) = p_c(\mathbf{C}|\hat{\theta}) \prod_{\hat{\theta}_k=0}(1 - P_d) \prod_{\hat{\theta}_k>0} P_d \cdot p(\mathbf{z}^{\hat{\theta}_k}|\mathbf{h}^k(\mathbf{x})), \tag{52}$$

where $\hat{\theta}$ is the most significant association hypothesis selected from all valid hypotheses by

$$\hat{\theta} = \arg\max_{\boldsymbol{\theta}} p(\mathbf{Z}|\mathbf{H_x}, \boldsymbol{\theta}). \tag{53}$$

By maximizing the logarithmic likelihood function, the target localization problem is converted into an NLS optimization problem as

$$\hat{\mathbf{x}} = \arg\min \sum_{\hat{\theta}_k>0} \left(\mathbf{z}^{\hat{\theta}_k} - \mathbf{h}^k(\mathbf{x})\right)^T \boldsymbol{\Sigma}^{-1} \left(\mathbf{z}^{\hat{\theta}_k} - \mathbf{h}^k(\mathbf{x})\right). \tag{54}$$

The NLS optimization problem can be solved by numerical methods initialized with the target location estimated in the first step.

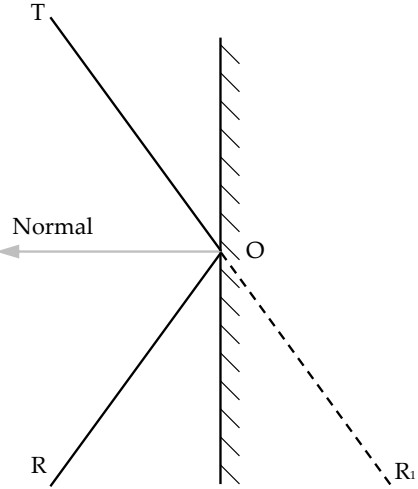

**Figure 3.** A single-bounce path from radar $R$ to the target $T$ reflected at position $O$ and the equivalent path from virtual radar $R_1$ to the target directly.

## 4. Simulation Results

Simulations are presented to evaluate the proposed algorithm in this section. The scenario setup is illustrated in Figure 4, where the dark blocks represent the obstacles in the scenario. A radar locates at $(60, 40)$ with the main lobe direction facing north. A total of 100 targets distributed from $(35.5, 125.5)$ to $(134.5, 125.5)$ along the x-axis direction at intervals of 1 m are simulated to study the robustness of the proposed localization algorithm in different multipath propagation conditions. The sampled locations computed offline are located at grids from $(0, 0)$ to $(160, 180)$ with intervals of 1 m along the x and y directions. Therefore, the off-grid error of localization caused by the distance between the targets and their nearest sample is $\sqrt{2}/2$ m.

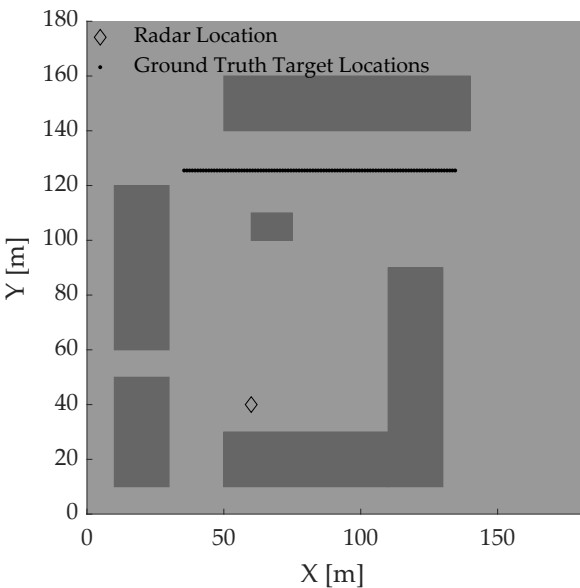

**Figure 4.** Simulation scenario setup.

Figure 5 illustrates the number of round-trip multipath components in the study area restricted by a rectangle with length of 160 m and width of 180 m, where the times of bounces for a simulated single-trip path in Figure 5a,b is limited to less than one and two, respectively. The increase in reflection times complicates the multipath propagation model not only in the maximum received multipath components but also in the heterogeneity of multipath components. As is shown in Figure 5, the maximum number of round-trip paths is 25 for the single-bounce case, while the number increases to 144 for the double-bounce case. Moreover, the propagation model varies quickly and becomes more sensitive to target locations as the reflection times increase. However, the increased paths do not contribute to the localization area because they still cannot reach the shadow regions compared to the single-bounce case. Considering the requirement of radar signal receiving energy, the times of bounces are limited to less than one in the following simulations, and single-trip reflection paths of more than one bounce are considered undetectable.

The ground truth of target multipath measurements as well as the cluttered measurements are illustrated in Figure 6. In the simulation, the probability of each path being detected $P_D$ is set to 0.8. Since the measurement of distance and angle are independent, zero-mean Gaussian noises are added to the distance and angle measurements separately with the corresponding standard deviation (STD) of the noises as $std(\tau)$ and $std(\phi)$. Therefore, the covariance of measurement is given as

$$\mathbf{\Sigma} = \begin{bmatrix} std^2(\tau) & 0 \\ 0 & std^2(\phi) \end{bmatrix}. \tag{55}$$

We assume the Poisson rate $\lambda$ equals 20, which means that each radar scan produces 20 cluttered measurements on average. Each cluttered measurement $\mathbf{c} = [\tau, \phi]$ is generated by two uniform distributions as

$$\begin{aligned} \tau &\sim U(0, 350), \\ \phi &\sim U(0, 180), \end{aligned} \tag{56}$$

where $[0, 350]$ and $[0, 180]$ are distance and angle range of radar, respectively.

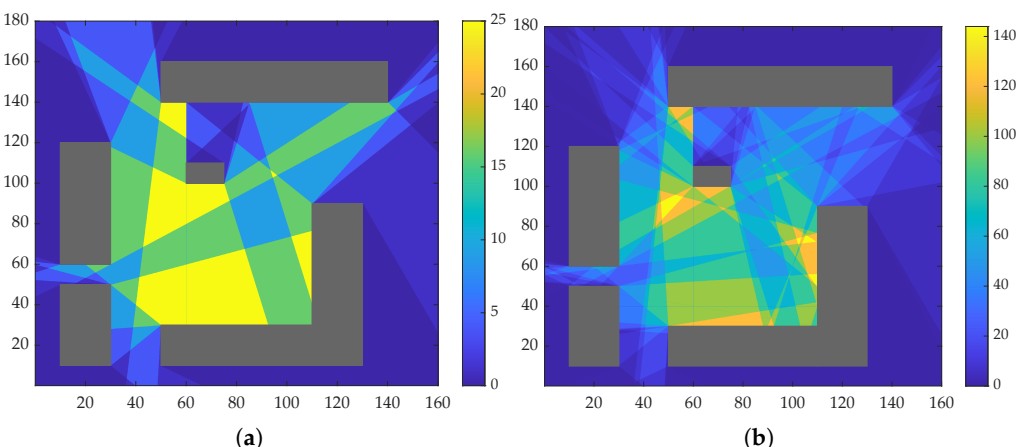

**Figure 5.** Number of round-trip multipaths in the study area: (**a**) Reflection is limited to single-bounce; (**b**) reflection is limited to double-bounce.

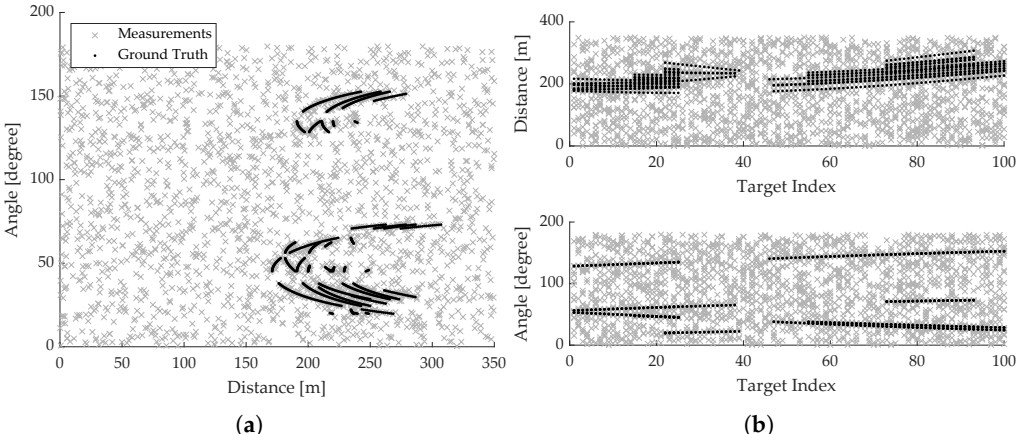

**Figure 6.** Ground truth and measurements with clutter: (**a**) target trajectory in the measurement space; (**b**) measurements versus target index.

The localization results of the proposed algorithm are illustrated in Figure 7, where the dotted lines denote the grid-based localization error of the first step, and the solid lines denote the NLS-based localization error of the second step. Results from index 38 to 48 are excluded because no path of the target behind the obstacle propagates to the radar as shown in Figure 6b. Localization error is defined as the Euclidean distance between the estimated target location and the ground truth by $\|\hat{\mathbf{x}} - \mathbf{x}\|$. The red lines present the simulation results that the STD of distance and angle measurement equals one. In the case of low noise level, the grid-based method results are accurate, and the off-grid error is the main cause of localization error. Therefore, the NLS-based method can improve the localization accuracy based on the correct estimation of the multipath propagation model. However, in the case of high noise levels displayed in the blue lines, the NLS-based approach shows little improvement due to the incorrect multipath propagation model estimated in the first step.

Furthermore, the impact of measurement noise is analyzed by Monte Carlo simulation of a target located at (110.5, 125.5). The root mean square error (RMSE) of localization is obtained by 100 simulations as

$$RMSE = \sqrt{\frac{1}{100}\sum_{i=1}^{100}\|\hat{\mathbf{x}}_i - \mathbf{x}\|^2}. \tag{57}$$

Figure 8 shows the RMSE with respect to different distance and angle noise STD, where Figure 8a is the grid-based result, and Figure 8b is the NLS-based result. The simulation result in Figure 8a shows that the first step of the proposed algorithm is more sensitive to distance measurement than angle measurement. Accurate localization results are obtained regardless of angle error in the condition that the STD of distance noise is less than one. However, the localization error increases along with the STD of distance when the STD of angle is less than one. The reason is that nearby samples share similar received angles, while the distance varies greatly. Therefore, if the distance measurement is precise, measurements with large angle errors are not associated with nearby samples. However, if the angle measurement is accurate and the distance measurement is inaccurate, the measurement is more likely to be associated with a nearby sample with similar angles and result in a wrong estimated location. Results illustrated in Figure 8b validate the effectiveness of the NLS-based re-localization process when the first step estimates the correct sample location.

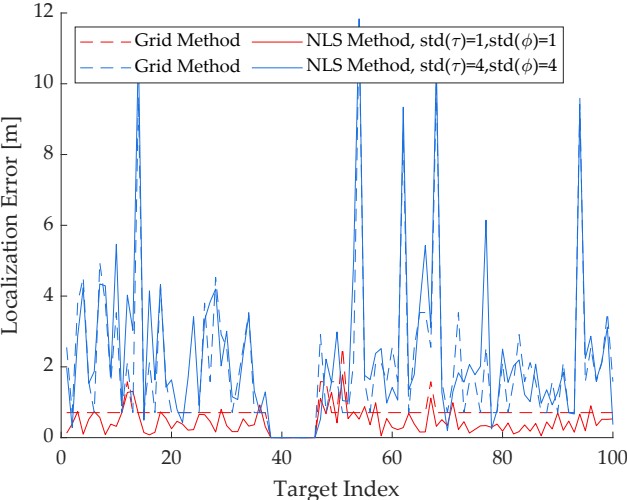

**Figure 7.** Localization error of the proposed algorithm at different locations.

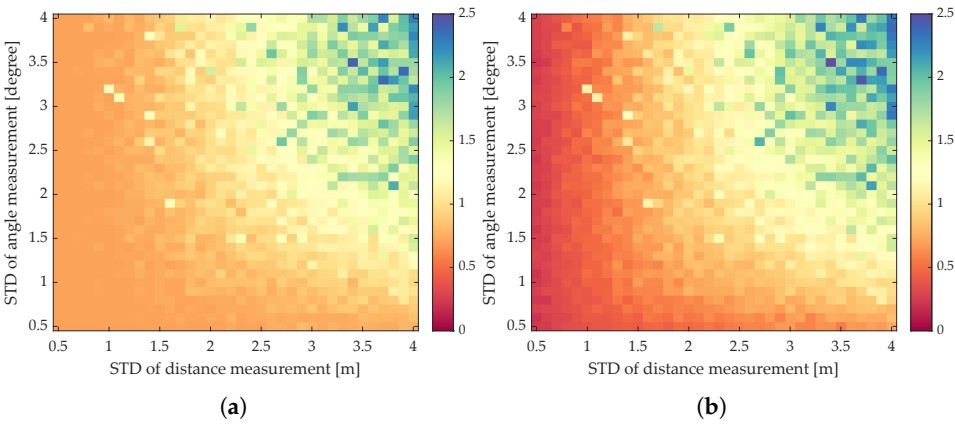

(a)　　　　　　　　　　　　　　　　　　　　　　　　(b)

**Figure 8.** RMSE of localization versus STD of the measurement: (**a**) grid-based results; (**b**) NLS-based results.

The detailed performance of the proposed algorithm is illustrated in Figure 9. The NLS-based method improves the localization accuracy when the STD of distance is less than 1.5. When the STD of the angle equals 0.5, the proposed algorithm can locate the target in the high clutter environment with precision less than the off-grid error. When the STD of the angle becomes larger, the proposed algorithm still has fairly good performance, although the NLS-based method has no more improvement. If the radar distance measurement is accurate while the angle measurement is inaccurate, which is often the case, the proposed algorithm provides precise location results as shown in Figure 9b.

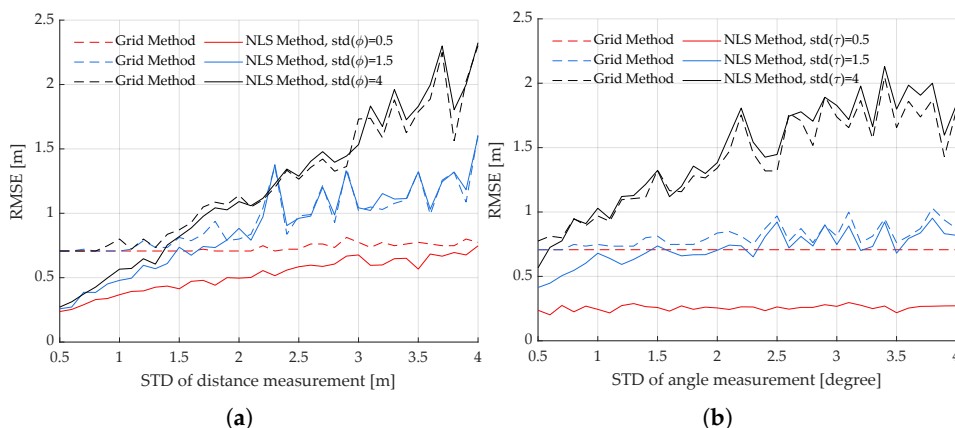

**Figure 9.** RMSE of localization versus STD of the measurement: (**a**) RMSE versus distance measurement; (**b**) RMSE versus angle measurement.

Existing multipath localization algorithms do not consider detections with clutter, and most of them have strict restrictions on the target scene, such as in the corner of a corridor [28] or in an urban canyon [25]. We choose an NLOS target localization algorithm based on grid matching of TOA returns proposed in [45] to verify the performance of the proposed algorithm. Since the original referenced algorithm does not consider measurements with clutter, the performance of the referenced algorithm is evaluated by Monte Carlo simulations with clean measurements as well as cluttered measurements. Since the referenced TOA-only algorithm does not use angle measurements, we restrict the proposed algorithm by fixed STD of angle measurements and analyze the performance of the two algorithms with varying STD of distance measurement. The Monte Carlo results illustrated in Figure 10 show that the proposed algorithm of our paper outperforms the referenced algorithm under both conditions even if the STD of angle measurement is set up to four. Table 1 shows the RMSE of the two localization algorithms. The proposed algorithm performs well in both scenes, and the RMSEs are less than 1 m. However, the performance of the referenced algorithm decreases significantly in the presence of cluttered measurement, and the algorithm is not applicable in dense clutter environments.

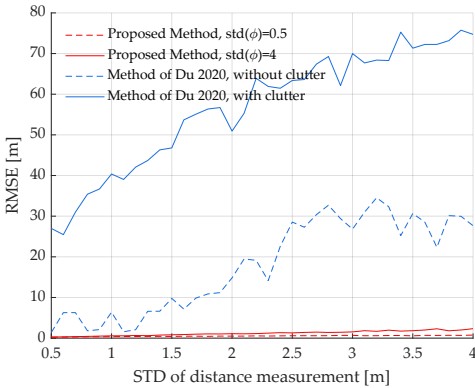

**Figure 10.** Comparison with the referenced algorithm in [45].

**Table 1.** RMSE of simulation results.

|  | **Proposed Algorithm** | | **Referenced Algorithm** | |
| --- | --- | --- | --- | --- |
|  | **std($\tau$) = 0.5** | **std($\tau$) = 1** | **std($\tau$) = 0.5** | **std($\tau$) = 1** |
| RMSE with no clutter (m) | 0.2247 | 0.3294 | 1.3969 | 6.3886 |
| RMSE with dense clutter (m) | 0.2370 | 0.3676 | 26.9842 | 40.3755 |

## 5. Conclusions

In this work, we presented a target localization method in multipath scenarios with high clutter. Multipath propagation is predicted by the ray-tracing technique based on prior knowledge of the urban environment. The proposed method determines the multipath propagation model by the likelihood function of multipath measurement and the predicted multipath with respect to all possible association hypotheses between them. Approximation of the proposed likelihood function is derived by excluding impossible hypotheses with a gating threshold. Then, accurate target location is obtained by an NLS optimization based on the estimated multipath propagation model. Simulation results showed that the proposed method provides robust and accurate estimation of target location in a high clutter environment. The promotion of the proposed multi-measurement likelihood function to the classic single-measurement tracking problem can be an interesting and practical topic. Particularly, considering the association explosion in the multi-target multi-measurement localization problem, integration of localization and the tracking process is expected to provide a possible solution in future work.

**Author Contributions:** Conceptualization, R.D. and Z.W.; data curation, R.D.; formal analysis, R.D.; funding acquisition, L.J.; investigation, R.D.; methodology, R.D.; project administration, L.J.; resources, Z.W.; software, R.D.; supervision, Z.W.; validation, R.D.; visualization, R.D.; writing—original draft, R.D.; writing—review and editing, S.Z. All authors have read and agreed to the published version of the manuscript.

**Funding:** This research was funded by National Defense Basic Research program of China grant number WDZC20195500208.

**Institutional Review Board Statement:** Not applicable.

**Informed Consent Statement:** Not applicable.

**Data Availability Statement:** Not applicable.

**Conflicts of Interest:** The authors declare no conflict of interest.

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
