# Peer review of "Radar Target Localization with Multipath Exploitation in Dense Clutter Environments"

_applsci, doi:10.3390/app13042032_

Round 1

Reviewer 1 Report

Dear authors, 

I have only minor comments with typos. See attached PDF.

Best regards

Author Response

Thanks for the comments.

  1. The grammatical and symbolic errors have been checked and corrected.
  2. Notation in Figure 4 has been revised.

Reviewer 2 Report

The overall paper presents a new method to detect targets with a higher precision in a "crowded" environment. However this is a theoretical approach. It would be interesting to present, maybe in a future paper, the measurements and comparisons with the new method from a real environment with the findings from the experiment.

Be careful to the page 5 (line 172): "3. method" it is not clear if there is a sub-chapter or a chapter... 

In page 11, at line 307 it is presented "AOA" but it is not mentioned what it is about... the reader could just guess or intuits...

It would be good to present also the figure with the graph that should show the STD of angle measurement [degree] also for the proposed method.

Author Response

Response to Reviewer 2 Comments:

Point 1: Be careful to the page 5 (line 172): "3. method" it is not clear if there is a sub-chapter or a chapter

Response 1: It is a chapter and the error typing has been revised.

Point 2: In page 11, at line 307 it is presented "AOA" but it is not mentioned what it is about... the reader could just guess or intuits

Response 2: Sorry for the mistake. I used angle-of-arrival (AOA) in the draft, and changed to direction-of-arrival (DOA). It was an omission and has been revised.

Point 3: It would be good to present also the figure with the graph that should show the STD of angle measurement [degree] also for the proposed method.

Response 3: Simulation results with regard to STD of angle measurement [degree] is illustrated in Figure9(b). In comparison with other methods, only range measurements were used so there is no change in STD of angle measurement in Figure 10.

Reviewer 3 Report

Line 78, 82 and other places in a manuscript missing a dash for “ray-tracing”. Should be unified, unless there is a semantic reason to keep this way.

Line 76. Since the proposed method is based on ray-tracing technique, a brief description (without any technical details) about ray-tracing might be included before starting to cite important works in line 76.

Abstract should have results included, and which model it outperforms.

There is something missing in First sentence of Chapter 2.1. in Line 104. I believe electromagnetics is plural. Also, Line 172, should in be capital letter. Space between units and metrics, Line 328. Further English proofreading is required.

Line 112. I understand that implementation of ray-tracing method might be out of the scope of this paper, but a reference on which it is based, software or a source code of implementation should be provided.

Line 117. Target position is defined as x, while in Line 130, samples are also defined as x. I suggest improving the notation.

A summary table with obtained results should be provided to make it easier to understand the paper.

Authors compare their results with a work from conference in 2020. As far as I understand, that work does not have any citation. It is unclear if this is the state-of-the-art. Results should be compared and discussed with more important works from the area.

Author Response

Response to Reviewer 3 Comments:

Thanks for your comments.

Point 1: Line 78, 82 and other places in a manuscript missing a dash for “ray-tracing”. Should be unified, unless there is a semantic reason to keep this way.

Response 1: All “ray tracing” have been unified as “ray-tracing”.

Point 2: Line 76. Since the proposed method is based on ray-tracing technique, a brief description (without any technical details) about ray-tracing might be included before starting to cite important works in line 76.

Response 2: Brief description of ray-tracing was given in Section 2.1 in the original manuscript. In the revised manuscript, we add a short description in the  line 76.

Point 3: Abstract should have results included, and which model it outperforms.

Response 3: We have revised the abstract and added simulation results in the abstract.

Point 4: There is something missing in First sentence of Chapter 2.1. in Line 104. I believe electromagnetics is plural. Also, Line 172, should in be capital letter. Space between units and metrics, Line 328. Further English proofreading is required.

Response 4: The grammatical and symbolic errors have been checked and corrected.

Point 5: Line 112. I understand that implementation of ray-tracing method might be out of the scope of this paper, but a reference on which it is based, software or a source code of implementation should be provided.

Response 5: In this paper, a commercial software Wireless Insite provided by REMCOM was used and we add a reference on it.

Point 6: Line 117. Target position is defined as x, while in Line 130, samples are also defined as x. I suggest improving the notation.

Response 6: Thanks for the advice. Notation for samples has been revised as s.

Point 7: A summary table with obtained results should be provided to make it easier to understand the paper.

Response 7:  A summary table has been added in line 408 to compare the two algorithms.

Point 8: Authors compare their results with a work from conference in 2020. As far as I understand, that work does not have any citation. It is unclear if this is the state-of-the-art. Results should be compared and discussed with more important works from the area.

Response 8: At present, there is no standard data set in the field of localization with multipath exploitation, and most of the papers do not provide their algorithm code. Therefore, I have implemented the code of some papers by myself, but it cannot be guaranteed to be completely consistent with the original author. And these algorithms have different applicable scenarios. Some of the multipath localization algorithms have strict restrictions on the target scene, such as in the corner of the corridor or in the urban canyon. Therefore, it is not convenient to compare such algorithm with the proposed algorithm. 

Reviewer 4 Report

In section "3. Method" has controversy: in line 173 written “two-stage localization algorithm is introduced” but in line 174 written “illustrates the framework of target detection and localization method”. There are disagreement with “algorithm” and “method”

Author Response

Point 1: In section "3. Method" has controversy: in line 173 written “two-stage localization algorithm is introduced” but in line 174 written “illustrates the framework of target detection and localization method”. There are disagreement with “algorithm” and “method”.

Response 1: The “method” has been unified as “algorithm”.

Round 2

Reviewer 3 Report

Authors addressed my comments.